# Vaccine Confidence During Public Health Challenges and Prior to HPV Vaccine Introduction in Mali

**DOI:** 10.3390/vaccines13050535

**Published:** 2025-05-17

**Authors:** Tiffani Crippin, Karamoko Tounkara, Ibrahima Diarra, Pierre Kamate, Sarah Beseme, Matthew Murphy, Hayley Munir, Amalle Keita Daou, Garan Dabo, Ibrahima Téguété, Ousmane Koita, Anne S. De Groot

**Affiliations:** 1GAIA Vaccine Foundation, Providence, RI 02909, USA; 2GAIA Vaccine Foundation, Bamako 999053, Mali; tounkara.karamoko@gmail.com; 3National Center for Immunization, Bamako 999053, Mali; idiarra50@yahoo.fr; 4Regional Department of Health (DRS), Bamako 999053, Mali; drpierrekamate@gmail.com; 5Warren Alpert School of Medicine, Brown University, Providence, RI 02903, USA; 6Criminal Justice Sciences Faculty, Illinois State University, Normal, IL 61761, USA; hrmunir@ilstu.edu; 7Clinique Dr Amalle Keita, Bamako 999053, Mali; 8COVID Treatment Unit, Hospital of Mali, Bamako 999053, Mali; 9Department of Obstetrics and Gynecology, Hospital Gabriel Touré, Bamako 999053, Mali; 10Laboratory of Molecular Biology (LBMA), Faculty of Arts & Sciences (FAST), University of Sciences, Techniques and Technologies of Bamako (USTTB), Bamako 999053, Mali; 11EpiVax, Inc., Providence, RI 02909, USA

**Keywords:** vaccination confidence, vaccination campaign, COVID-19, HPV, Mali, West Africa

## Abstract

**Background/Objectives**: Public health activities and the roll-out of new vaccines such as the HPV vaccine in Mali have been disrupted by both the COVID-19 pandemic and by political unrest from March 2020 until recently. The HPV vaccine was introduced into the childhood vaccine program in 2024. Given the persistent threat of ongoing infectious disease epidemics, there is a pressing need to understand the factors influencing vaccine uptake in Mali and other low- and middle-income countries. **Methods**: To address this need, the GAIA Vaccine Foundation (GAIA VF), a nongovernmental organization (NGO), conducted a study of vaccine confidence in Bamako, the country’s capital and primary population center, between September 2021 and March 2022 at 12 community based primary care clinics and 1 rural primary care clinic. The study was coupled with a vaccine outreach and education campaign at each site. **Results**: Study staff collected information on vaccine confidence in surveys from 3445 community participants. Healthcare providers were also recruited from the 13 sites for vaccine-related training, and 140 of these participants completed pre- and post-surveys on their vaccine knowledge and confidence. Survey results indicate that community members trust their primary care providers more than they trust the government. However, primary care providers trust government sources of information more than other sources for guidance on vaccines. **Conclusions**: As new vaccines are introduced, engaging key healthcare leaders to inform healthcare providers and developing primary care provider-led community outreach will be critically important to gaining community confidence prior to and during vaccination campaigns in the future.

## 1. Introduction

At the end of 2023, Mali reported 33,162 confirmed cases of COVID-19 (Coronavirus Disease) and 743 deaths to the World Health Organization, falling roughly into the regional average [1]. Unfortunately, since accurate counts of cases depend on regular testing, those figures are unlikely to be accurate, as Mali struggled with testing since the start of the pandemic [2]. Serological surveys conducted in Bamako during the first wave of the epidemic also demonstrated that SARS-CoV-2 (Severe Acute Respiratory Syndrome Coronavirus 2) exposure was more widespread than had been reported [3,4].

The period of COVID vaccine introduction was turbulent from a political perspective, which may have had an impact on attitudes toward vaccines in general and the COVID vaccine in particular. The first doses of COVID-19 vaccines arrived in Mali in March 2021 under the civilian government installed by the first coup d’état. That government would be replaced by a military dictatorship two months later. As of November 2023, only 22% of the population had received a single dose of a COVID vaccine, with only 18% completing a primary series [5]. Despite the efforts of public health officials, Mali had the 13th worst COVID vaccination rate in the world [1].

The US-based GAIA Vaccine Foundation (GAIA VF) has a long history of HIV (Human Immunodeficiency Virus), TB (Tuberculosis), and HPV (Human Papillomavirus) vaccine-focused research in Mali [6,7,8,9]. Given the opportunity to evaluate the challenges involved in introducing a new vaccine for a new virus (SARS-CoV-2) in September 2021 and the impending rollout of the HPV vaccine, GAIA VF launched a survey-based vaccine confidence research project in two areas of Mali. Twelve community health clinics and their coordinating institution, the Center of Reference in Commune I of Bamako, participated along with one rural community clinic in the village of Keniero, about 70 km outside of Bamako in the district of Siby. The goal of the vaccine confidence project was to characterize the impact of the introduction of a new vaccine amid a pandemic and (ongoing) political instability on community and healthcare providers’ (HCPs) knowledge, attitudes, and vaccine confidence. The survey project was coupled with a two-pronged vaccine promotion intervention that included HCP trainings and a public education campaign building upon GAIA VF’s previous work around promoting vaccine uptake.

During the same period of time, a country-wide campaign to vaccinate against cervical cancer was in preparation. In a previous study, we evaluated attitudes and interest in HPV vaccination and learned that vaccine acceptability was high in Bamako [7]. We were aware that GAVI would begin supplying the HPV vaccine to Mali in the fall of 2024 and that the vaccine would be given to 10-year-old girls as part of the routine childhood vaccine program. We therefore reassessed attitudes and behaviors regarding HPV vaccines as part of the vaccine confidence survey.

In this manuscript, we present lessons learned from COVID vaccine outreach that will inform the approach to future vaccines and the results of surveys of both community members and HCPs. We also provide insights as to HPV vaccine acceptability immediately prior to the introduction of the HPV vaccine in Mali. This study’s results fill an important knowledge gap related to vaccine usage in Mali as well as the impact of political instability and a change from democratic to nondemocratic governance on the public health response to vaccination. Methods developed to address barriers to COVID-19 vaccine uptake will provide important insights into effective vaccine interventions in Mali and other countries faced with similar challenges to vaccine confidence.

## 2. Materials and Methods

Building upon firsthand experience from a pilot HPV vaccination program, GAIA VF developed the components of both a community and HCP vaccine promotion intervention in collaboration with Malian physicians, stakeholders, and community leaders to support COVID-19 vaccine rollout between February and June 2021. GAIA VF staff engaged with Malian public health officials to address vaccination plans in Bamako. The team identified 12 community clinics (the first level of health structures in Mali) with qualified staff in Bamako and 1 community clinic in rural Keniero for participation in the project (Appendix A—Map). With support from Malian public health authorities, we developed materials (cloth, posters, surveys, and radio advertisements) that were used to educate the public about the COVID-19 vaccine and the vaccine confidence program.

### 2.1. Public Education Intervention on Vaccines

Community healthcare workers (CHWs) led the public education intervention about the vaccine in each of the 13 participating communities. The CHWs recruited for this project were primarily women who belonged to associations that regularly perform community outreach for their local clinic. Each participating CHW was trained in classroom-style sessions under the direction of members of the Regional Health Directorate and the National Center for Immunization. Twice per week, the CHWs set up events in an open area in their community to educate the public about vaccines and other measures for preventing infectious diseases such as COVID-19, HIV, and HPV.

Materials for the public education intervention included the following: (1) two posters, (2) radio announcements, (3) a storytelling cloth design that adapted from prior GAIA VF organized vaccine promotion campaigns and produced by a local textile plant (Figure 1), and (4) a survey on COVID-19 vaccine attitudes and beliefs. Radio announcements began in September 2021 to advertise the intervention (a sample of the dialogue is provided in the Appendix A). The posters provided descriptions of how COVID-19 is transmitted and instructions on how to protect yourself from contracting the virus in both French and Bambara, along with illustrations. The posters were displayed at each of the clinics and were publicly visible throughout the intervention (Appendix A).

Figure 1 of the storytelling cloth illustrates public health messages that were being reinforced during the training and outreach program. The key elements are “social distancing” (tape measure), “masking if infected” (face mask), “Handwashing” (hand with water drops), and “COVID-19 is transmitted from person to person” (round figure with the individual infection rate illustrated as an expanding number of points emanating from the central figure). This cloth was distributed to members of the outreach team in colors selected by team members, and the outreach workers, clinicians, and project leaders wore the cloth during the outreach campaign.

The storytelling cloth is a communication method that GAIA VF developed and used in a previous prevention/education campaign to inform the community about HPV and the risks of cervical cancer [6]. The storytelling cloth acts as a unifying storyboard for CHWs and HCPs during the public education intervention. The fabric was distributed to both CHWs and HCPs and was used to sew outfits, handbags, and other wearable items (Appendix A). CHWs wore the storytelling cloth during education events as a tool to explain the message, and HCPs wore the cloth at the clinic while surveying participants.

Working with officials from the local and regional Department of Health, GAIA VF team members developed a survey to assess the impact of the intervention, to understand perceptions of COVID-19 and its vaccines, to capture sources of trusted information in the community, and to evaluate the impact of misinformation [10]. For additional details on the drafting process, see Appendix A, where additional information about the IRB approval and copies of the surveys are available.

Public education intervention activities began on 1 October 2021, after COVID-19 vaccines became more widely available in Mali. The HPV vaccine was not yet available; however, due to the impending implementation of national HPV vaccination, cervical cancer and HPV vaccination were included in the outreach education topics. Visits to the 13 community clinics participating in the project confirmed that COVID-19 vaccines were available at no cost at each site.

While the public education intervention was ongoing in the community, trained nurses wearing the storytelling cloth orally surveyed community members who visited their clinic about their sources of information about vaccines, vaccine practices in their own family, and their confidence in vaccine safety. They asked any adult over 18 to participate, and upon receiving verbal consent, administered the survey to each participant individually. Questions about the COVID-19 vaccine and the HPV vaccine were included in the survey. Consenting participants were surveyed regardless of whether they had participated in a public education intervention activity or not. Given the high rates of illiteracy, literacy was not a prerequisite for study participation. Consent was given by thumbprint or signature, and survey responses were obtained orally for all participants. Participants were able to withdraw from study participation at any time and were informed that they could skip any question they did not want to answer.

### 2.2. Healthcare Provider Training

In addition to the multicomponent public education intervention, this study also captured the impact of targeted training for HealthCare Providers (HCPs). A total of 70 HCPs participated in specialized infectious disease training hosted by 4 infectious disease specialists from the Ministry of Health and the principal hospital in Bamako. Topics included COVID-19 vaccines, childhood vaccines, cervical cancer, and HPV vaccines. Each presenter developed their own curriculum and slideshows and led their portion of the training events. A photo of the training is available as Appendix A.

GAIA VF developed a survey to assess whether this supplemental, targeted training was linked with increased HCP knowledge and confidence in vaccines. Pre- and post-training surveys were administered to participants who gave consent (see Appendix A for information about the drafting process, human studies approval, consent procedures, and copies of the surveys). In total, 70 healthcare providers participated in the training, and 140 surveys were collected.

### 2.3. Data Inputs and Analysis

The National Center for Immunization provided Malian COVID-19 vaccination data. That data were disaggregated using Commune for a comparative analysis of study sites versus non-study sites. A project manager in Mali input the community survey data using Google Forms. GAIA’s US-based team exported that data into an Excel spreadsheet for analysis. The US team also coded the HCP surveys directly into a spreadsheet and conducted descriptive and comparative analyses using maximum likelihood estimation techniques to determine statistical significance. Missing data were dropped from all analyses where that question/data point was included. When evaluating the pre-/post-surveys for HCP, the HCP would be dropped from the analysis entirely if they did not answer the question being evaluated.

## 3. Results

A total of 3445 community members and 140 healthcare providers participated in the study. Demographic data for all participants are included in Appendix A. Most community participants were female, had children, and had little to no formal education. Participating HCPs were primarily female midwives and nurses. A key finding was that community outreach was essential for improving vaccine confidence. As shown in Appendix A, most of the community outreach participants were more confident about vaccines following their participation in the intervention.

### 3.1. Public Education Intervention

A total of 3445 surveys were collected during the outreach events. The vaccine confidence questions focused on the vaccines used during the pandemic (COVID-19) and the COVID-19 and HPV vaccines. Most community participants had heard of COVID-19 (87%; N = 2986/3445) and agreed it was a major health risk (73%; N = 2511/3445), but only 54% (N = 1849/3445) said that they knew what caused COVID-19 transmission. When asked if they followed COVID-19 prevention guidelines (i.e., mask wearing, handwashing, and social distancing), about 68% (N = 2330/3445) said that they did. Only 24% (N = 833/3445) of respondents had ever been tested for COVID-19, and only 21% (N = 736/3445) were vaccinated. A total of 56% of unvaccinated respondents said they would agree to be vaccinated. A lack of confidence in the vaccine was a major reason for avoiding vaccination: those respondents who were unwilling to be vaccinated reported “I am worried about side effects” (47%; N = 209/445) or “I don’t think it will be prudent” (25%; 110/445). Table 1 shows that most respondents reported hearing misinformation about COVID-19 and/or COVID-19 vaccines. A smaller proportion of the population (between 2% and 19%) believed the rumors they heard were true or possible.

Table 2 shows the odds ratios from three logistic regression models estimating the relationship between relevant independent variables and COVID-19 testing and vaccination. Education level was a strong predictor of prior testing, vaccination, and willingness to be vaccinated for COVID-19. A 1-unit increase in education led to a 24.5% increase in the odds of being previously tested, a 36.8% increase in the odds of being previously vaccinated, and a 20.1% increase in the odds of being willing to be vaccinated. Older individuals were more likely to have been vaccinated and be willing to accept vaccination, although age was not a statistically significant predictor of testing. Gender and religion both appeared to play a role in vaccine uptake. Men were 40% more likely to have been previously vaccinated than women were, but sex had no statistically significant relationship with prior testing or willingness to be vaccinated. Finally, a 1-unit increase in religiosity increased the odds of being previously tested and vaccinated by 9.1% and 20.5%, respectively, but decreased the odds of being willing to be vaccinated by 1.2%.

The survey also included a series of questions asking how often the respondent trusted various groups for vaccination advice. Local healthcare providers were a major source of trusted information. There was a striking difference between the numbers of community-based participants who always trusted their community healthcare provider (2165) versus the government health authorities (976). Similarly, only 127 individuals said that they never trust vaccination advice from their community health provider, while 633 never accept the advice of government health authorities. The full table of responses is included in Appendix A.

Table 3 presents the odds ratios from two logistic regression models estimating the relationship between relevant independent variables and a respondent’s trust in the vaccination advice of their local community clinic HCPs and government health authorities. Men were 36% less likely to trust the advice of their local HCP compared to women, but there was no distinguishable difference between men and women when it came to trusting government health authorities. Education level, exposure to a HCP discussion, public health event, and radio announcement all predict both trust in local HCPs and government health authorities. A 1-unit increase in education increased the odds of trusting one’s local HCP for vaccination advice by 41.3% and the odds of trusting the government health authorities by 11.1%. Being exposed to a public education event increased the odds of trusting local HCP by 51.4% and the odds of trusting the government health authorities by 21.5%. Exposure to the “story-telling” fabric was positively correlated with trust in government health authorities, but not local HCPs. Exposure to a public health event and exposure to the fabric are highly correlated, leading to the possibility of multicollinearity issues in the model.

### 3.2. Healthcare Provider Training Intervention

Like community participants, healthcare providers attending a training event were asked specific questions about COVID-19. There was little to no difference in their responses before and after each training. Generally, about 79% (111/140) of healthcare providers had received the COVID-19 vaccine, and 68% (95/140) of the surveyed healthcare providers knew someone (a family member, friend, or patient) who had been seriously ill or died from COVID-19. Most healthcare providers agreed that COVID-19 was a major public health threat, still existed in Mali, survives in hot climates, and can be serious, but only 61% (85/140) believed that the official COVID-19 figures reflected the reality of the disease in Mali. The full responses to COVID-19 questions asked of healthcare providers are included in Appendix A.

The percentage of HCPs who strongly agree that “Vaccines are safe”, “Vaccine are effective”, and “Vaccines are important for children” increased in the survey given after the training compared to the survey before the training. Trust in government health recommendations also increased in the post-educational session survey. These differences were all statistically significant with *p*-values below 0.05 (vaccines are safe, *p* < 0.001; vaccines are important for children, *p* < 0.01; vaccines are effective, *p* < 0.05; I trust the government for healthcare recommendations, *p* < 0.05). The full results are in Appendix A. Vaccine knowledge also increased from the pre- to post-survey. The survey contained a series of knowledge questions asking HCPs to match 13 childhood vaccines to the illnesses they help prevent and to identify the type of vaccines available for HPV and COVID-19, for a total of 15 questions. Before the training, HCPs answered 8.8 (59%) of them correctly; after the training, they answered 10.3 (69%) correctly. This difference is statistically significant, with *p* < 0.05.

When asked whom they trusted for vaccination advice, most local healthcare providers trusted the government health authorities (54%; N = 76/140), with only a single HCP indicating that they do not trust government health recommendations (N = 1/70). Other common sources of trusted information about vaccination included the local healthcare provider’s supervisor (24%; N = 33/140), family and friends (17%; N = 24/140), professors/scientists (12%; N = 17/140), and local political representatives (11%; N = 16/140).

## 4. Discussion

One of the most important findings from this research is that a successful vaccination campaign must be led by trusted sources of information. Previous studies demonstrated that local healthcare providers were the key to increasing vaccine confidence among community members [6]. This study provides even more evidence supporting that observation, as far more respondents (63%; N = 2165/3445) said that they always trust the recommendations of their community HCP (e.g., doctors, nurses, and midwives that run their local clinic). Given the level of inconsistency in government leadership during the pandemic, it is not surprising that so few people trusted the government as a source of health information compared to their local community healthcare providers [11]. This may be related to changes at the level of the presidency and the Ministry of Health during the course of the pandemic, while local doctors largely remained the same.

The data also show a gender and education bias in the vaccination rate. Men and individuals with more years of education were more likely to get vaccinated. Interestingly, data from the DRS show that Commune I, where this study took place, has the lowest vaccination rates in all of Bamako. The low rate of vaccination may be related, in part, to vaccination access. Others have experimented with bringing the vaccines into the community, a strategy that may work in Mali as well [12].

The predicted probability of being tested for or vaccinated against COVID-19 was about 1.5 and 3 points higher for individuals with higher levels of education than those with no education. Other studies have reported a correlation between low vaccine confidence in low-income, marginalized populations [13,14]. Community members living in Commune I in Bamako are the poorest, least educated residents living in the city of Bamako. Indeed, 25% (862/3410) of our survey participants had no formal schooling at all.

However, there was no correlation between education level and knowledge about COVID-19. Those with no reported formal education were just as likely as those with a high school-level education to have heard of the virus, consider it a major health risk, know its causes, and practice safety guidelines. This is perhaps due to the use of a multicomponent intervention design, which included educational materials that did not require the ability to read or write. As the local language, Bambara, is more commonly spoken than written, low literacy poses a particular challenge for public health outreach in Mali. The World Bank reports that only 22.1% of women and 40.4% of men can read [15]. In comparison, the region of Sub-Saharan Africa has a female literacy rate of 61.4% and a male literacy rate of 74.2%. The storytelling cloth and public education intervention combination ensured that even community members with no schooling at all could access important health information.

The findings also underscore the importance of training for HCPs. In this study, HCP knowledge increased meaningfully after exposure to training. Given community members’ trust in HCPs and lack of trust in the government, efforts to train HCPs likely play a significant role in vaccine confidence and vaccination rates. For healthcare providers themselves, knowing about vaccines and being trained on new vaccines was an important indicator of vaccine confidence, as has been shown by more developed countries [16]. Healthcare providers in our study were more likely to report having been vaccinated and being willing to vaccinate their children. A study in Massachusetts showed a similar link between HCPs and vaccination confidence for COVID and HPV [17]. However, among HCPs in developed countries, concerns over the risk of side effects associated with vaccines, preference for natural immunity, not having any chronic disease risk factors, and vaccine mandates are among factors cited as key barriers to getting vaccinated [17].

Compared to other studies on vaccine confidence, this study confirms that trusting the national public health agency would be strongly correlated with several vaccine outcomes [18]. We also note that other studies evaluated healthcare “connectedness” as a measure of vaccine confidence. For example, having preventive health services in the last year was associated with COVID-19 vaccination in Italy, Mexico, the United Kingdom (UK), and the United States (US), as well as vaccine confidence in the US (RR 1.10) [18]. Similar to the findings in our study, “trust in healthcare provider” and health system engagement were associated with routine vaccinations (although not necessarily with COVID-19 vaccination) and with vaccine uptake [19].

The timing of and access to vaccination may also be factors: in many countries, HCPs were prioritized, and in some parts of the USA, COVID-19 vaccination was mandated, resulting in high vaccination rates among HCPs. In Mali, HCPs were also prioritized as soon as vaccines became available. Access to medical centers has been associated with higher HPV vaccination rates [20]. A review of vaccine confidence studies showed that little information is available on “convenience”, and this should be included in future studies of vaccine confidence. For example, it is not clear whether “access to vaccines” was responsible for the higher rate of vaccination among males. This was observed in Kenya as well [12].

As with the location of the vaccination program, integration of the program into routine vaccination is likely to be a key factor that improves vaccine acceptance [21]. Notably, Mali decided against providing the HPV vaccine at school and to both genders, factors that may decrease HPV vaccine uptake. Instead, the vaccine will be provided as a single dose and as part of the routine childhood vaccination schedule, factors that have been associated with increased vaccine uptake [21,22].

The survey results also showed that many well-known misinformation themes had emerged based on videos that were widely circulated on WhatsApp and other social media platforms. Misinformation has also been reported to have a negative effect on COVID-19 prevention measures in other LMIC (low- to middle-income countries) in West Africa, such as Cameroon and Ghana [23,24]. There was no correlation in this study between education level and belief in common rumors/misinformation about the COVID-19 vaccine, suggesting that the well-educated were equally exposed to misinformation as those with no formal schooling.

Thus, the finding that willingness to take a COVID-19 or HPV vaccine remains considerably higher in Mali than in the United States and Russia is encouraging. However, we also find that vaccine confidence has declined, compared to previous studies in the same location (Appendix A), which may be due to the importation of misinformation during the COVID-19 pandemic. Mali shares social media platforms with francophone countries such as France, which was a major source of misinformation during the pandemic [25]. In high-income countries, parents who fail to vaccinate their children do so knowing that they have access to high-quality care should their child become sick [18]. Improving access to vaccines and educating HCPs may improve acceptance rates for vaccines in LMIC; however, challenges may arise as access to misinformation also expands [26].

## 5. Conclusions

There are two main conclusions to be drawn from this study. First, community members, especially women, seem to trust the advice of their local clinic healthcare practitioners more than the official government healthcare authorities. Thus, vaccine campaigns should be led by informed (vaccine-educated) local healthcare providers. Second, improving access to vaccines should also be a priority for Mali, as men were 40% more likely to get vaccinated. One approach that has worked in the past is to conduct vaccinations in the community at sites where women and children are located.

Discussions with their local healthcare providers were the single most effective element of this campaign at increasing vaccine confidence, followed closely by public education intervention. Actively training front-line providers about vaccines (especially new vaccines such as the COVID-19 vaccine and the HPV vaccine) remains a high-impact activity that is likely to improve vaccine confidence overall. Fortunately, community-level healthcare practitioners trust government health authorities since they are typically the first stop for new healthcare information. Second, public education intervention appears to be effective in increasing knowledge and vaccine confidence. Maintaining the vaccine confidence of healthcare providers will clearly have an impact on vaccine confidence among these individuals, and this is also likely to have a positive impact on vaccine uptake in the general population. Furthermore, as Mali prepares for HPV vaccination campaigns, an investment in vaccine education prior to the mass vaccination campaign will surely have a significant impact on future generations of the people of Mali.

As with all research, there are several limitations to this study that should be improved upon in future research. First, all interviews were conducted by healthcare providers at community clinics. While this strategy was convenient and helped ensure that community members were comfortable, it may have introduced bias when community members were asked to identify whom they trusted for advice. Future projects should work to identify surveyors who are a part of the community, but not necessarily healthcare providers themselves. Second, this study was conducted in the poorest commune in Bamako and one rural site without the inclusion of any wealthier areas for comparison. Future studies would benefit from including a more well-educated and wealthy area for comparison. Finally, the survey for this project proved to be much too long, as we found many participants would stop answering after the first two pages. Limiting the number of questions is important to ensure the participant really gives themselves time to answer thoughtfully.

We plan to use the “story telling cloth” approach to support the HPV vaccine roll-out in Mali that is ongoing [27]. We will also evaluate aspects of vaccine confidence that have not previously been addressed in LMIC, such as access to vaccines (distance and availability) and healthcare (for lower-income individuals) as determinants of vaccination rates. We will continue to monitor the erosion of vaccine confidence in Mali and the impact of vaccine misinformation on the uptake of new vaccines.

## Figures and Tables

**Figure 1 vaccines-13-00535-f001:**
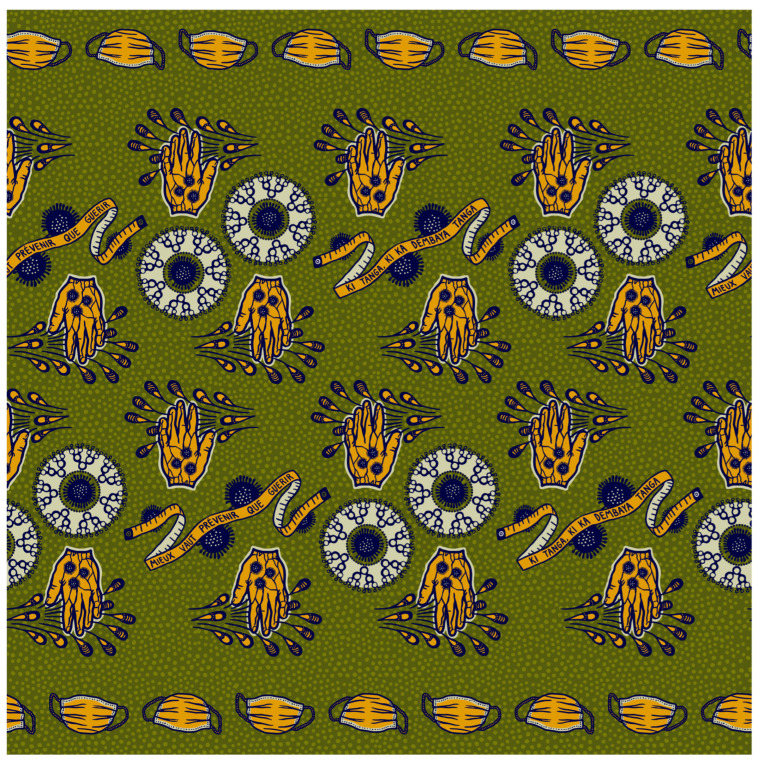
Story-telling cloth for the Corona Kele outreach workers.

**Table 1 vaccines-13-00535-t001:** COVID-19 vaccine rumors.

*Have You Heard These Rumors About COVID-19 Vaccines?*					
	Vaccine Contains Microchips	Vaccine Contains Magnets	Vaccine Will Change My DNA	Vaccine Will Cause COVID-19	Vaccine Will Turn Me into a Zombie
Never heard	34%	35%	33%	26%	37%
Heard but do not believe	45%	47%	48%	52%	49%
Heard and thinks it is possible	16%	15%	17%	19%	12%
Heard and believe	4%	3%	2%	3%	2%

**Table 2 vaccines-13-00535-t002:** Odd ratios for COVID-19 testing and vaccination.

	Prior COVID-19 Testing	Prior COVID-19 Vaccination	Willingness to Be Vaccinated
Sex	1.099 (0.129)	1.399 (0.158) ** [1.03, 1.91]	0.882 (0.108)
Age	1.004 (0.005)	1.037 (0.005) *** [1.03, 1.05]	1.018 (0.005) *** [1.01, 1.03]
Children	1.300 (0.172) *	1.029 (0.135)	0.864 (0.112)
Education	1.245 (0.044) *** [1.14, 1.36]	1.368 (0.049) *** [1.25, 1.51]	1.201 (0.044) *** [1.10, 1.31]
Religiosity	1.091 (0.052) * [0.99, 1.21]	1.205 (0.060) *** [1.07, 1.36]	0.880 (0.041) ** [0.81, 0.95]
Exposed to HCP Discussion			1.058 (0.057)
Exposed to Fabric			1.036 (0.050)
Exposed to Public Education Event			1.237 (0.065) *** [1.09, 1.41]
Exposed to Poster			1.015 (0.048)
Exposed to Radio Advertisement			1.223 (0.059) *** [1.09, 1.37]
Constant	0.155 (0.037) ***	0.035 (0.009) ***	0.382 (0.109) ***
Number of Observations	2957	3117	2427
Psuedo R^2^	0.066	0.190	0.123

Notes: * *p* < 0.1 ** *p* < 0.05 *** *p* < 0.01. All dependent variables were coded 0/1 for no/yes. Fixed effects by clinic included but not presented. Standard error in parentheses under estimates. The 95% confidence intervals for statistically significant estimates are shown in brackets under standard errors. Coding was as follows: Sex: 0 = female; 1 = male. Children: 0 = none; 1 = have children. Education 0 = no schooling; 1 = primary schooling; 2 = secondary schooling; 3 = tertiary schooling; 4 = university. Religiosity: 0 = never attend services; 1 = attend services for holidays; 2 = attend services monthly; 3= attend services weekly; 4 = attend services daily. Exposure 0 = none; 1 = exposed.

**Table 3 vaccines-13-00535-t003:** Odds ratios for whom subjects trust for vaccination advice.

	Trust Community HCPs	Trust Government Health Authorities
Sex	0.642 (0.161) * [0.47, 0.87]	1.029 (0.160)
Age	1.016 (0.012)	1.003 (0.006)
Children	1.229 (0.341)	1.205 (0.195)
Education	1.413 (0.124) *** [1.11, 1.79]	1.111 (0.051) ** [1.01, 1.22]
Religiosity	0.853 (0.089) * [0.71, 1.03]	1.027 (0.058)
Exposed to HCP Discussion	1.749 (0.183) *** [1.23, 2.49]	1.293 (0.083) *** [1.10, 1.52]
Exposed to Fabric	0.969 (0.108)	1.280 (0.077) *** [1.10, 1.49]
Exposed to Public Education Event	1.514 (0.163) *** [1.10, 2.09]	1.215 (0.081) *** [1.04, 1.42]
Exposed to Poster	0.904 (0.100)	0.972 (0.056)
Exposed to Radio Advertisement	1.478 (0.154) *** [1.09, 2.01]	1.422 (0.085) *** [1.22, 1.66]
Constant	1.064 (0.595)	0.797 (0.279)
Number of Observations	2966	2959
Psuedo R^2^	0.260	0.343

Notes: * *p* < 0.1 ** *p* < 0.05 *** *p* < 0.01. All dependent variableswere coded 0/1 for no/yes. Fixed effects by clinic were included but not presented. Standard error is included in parentheses under estimates. The 95% confidence intervals for statistically significant estimates are in brackets under standard errors. Coding was as follows: Sex: 0 = female; 1 = male. Children: 0 = none; 1 = have children. Education 0 = no schooling; 1 = primary schooling; 2 = secondary schooling; 3 = tertiary schooling; 4 = university. Religiosity: 0 = never attend services; 1 = attend services for holidays; 2 = attend services monthly; 3= attend services weekly; 4 = attend services daily. Exposure 0 = none 1 = exposed.

## Data Availability

The raw data for this study were collected in Google Forms. Access to the raw data is available upon request (TCrippin@GAIAVaccine.org). Other materials are provided in the Appendix A for this publication.

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
