# Peer review of "Vaccine Confidence During Public Health Challenges and Prior to HPV Vaccine Introduction in Mali"

_vaccines, 2025, doi:10.3390/vaccines13050535_

Round 1

Reviewer 1 Report

Comments and Suggestions for Authors

This study is an important research paper on vaccine confidence aimed at improving vaccination rates. It is a valuable study that considers public health challenges based on its findings.

1. The survey was conducted between September 2021 and March 2022, during the COVID-19 pandemic, and outlines the challenges associated with introducing a new vaccine. However, the content of the survey focuses on awareness of COVID-19 and COVID-19 vaccines, rather than awareness of cervical cancer and HPV vaccines. Therefore, the issues surrounding the introduction of the HPV vaccine are considered distinct from those discussed in this study.

2. There appears to be a discrepancy between the title and the content of the paper. It may be advisable to modify the title to better reflect the study’s focus for clarity. Additionally, it is recommended to adjust the citation format (reference format) in accordance with the submission guidelines.

Author Response

Reviewer 1

This study is an important research paper on vaccine confidence aimed at improving vaccination rates. It is a valuable study that considers public health challenges based on its findings.

Comment: The survey was conducted between September 2021 and March 2022, during the COVID-19 pandemic, and outlines the challenges associated with introducing a new vaccine. However, the content of the survey focuses on awareness of COVID-19 and COVID-19 vaccines, rather than awareness of cervical cancer and HPV vaccines. Therefore, the issues surrounding the introduction of the HPV vaccine are considered distinct from those discussed in this study.

Reply:

Thank you for the suggestions for clarification of the goals of this manuscript. It was indeed primarily planned during COVID but as we knew the HPV vaccine was soon to be introduced, we took this opportunity to evaluate the potential impact of the introduction of a new, distinctly different, vaccine for a new pathogen, and plan to apply any lessons learned to the ensuing HPV campaign.

We have clarified this purpose and where needed, added sections relating to COVID vaccine or HPV in the manuscript as also requested by other reviewers. (Changes are tracked. See: Abstract, Lines 76-81, 80-85, 89-90, 151-152, 161,172-173, At the time the survey was designed, the HPV vaccine was “going to be introduced” (it took much longer than anticipated). Fortunately, it was introduced soon after COVID. We have added several comments and text that elucidates this chronology (See for example Discussion Line 417 - close). 

Regarding the survey, it was designed to assess vaccine confidence overall during a period of public health challenges (COVID, and a military coup). However, due to the particularities of misinformation related to COVID, many of the questions (and results) were vaccine-specific. We have clarified this in the text (see previous reply) and included the surveys where both COVID and HPV questions are identifiable.

Specifically, as also requested by another reviewer, we have now included a short forward-looking section that addresses how we are integrating this information into our current HPV vaccine campaign activities (Line 417). We hope that this addition will clarify and improve the manuscript.

See new text in Discussion, and Conclusion (provided below).

Discussion

...Thus, the finding that willingness to take a COVID-19 or HPV vaccine remains considerably higher in Mali than in the United States and Russia is encouraging. However, we find that vaccine confidence has declined, compared to previous studies in the same location, which may due to the importation of misinformation during COVID. Mali shares social media platforms with francophone countries such as France, which was a major source of misinformation during the COVID pandemic (ADG, KT, personal observation). In the high income countries, parents who fail to vaccinate their children do so, knowing that they have access to high quality care should their child become sick. Improving access to vaccines and educating HCP may improve acceptance rates for vaccines in LMIC, however, challenges may arise, as access to misinformation also expands [see Solis et al]. . .

Also:

Conclusion.

. . . Discussions with their local healthcare providers was the single most effective element of this campaign at increasing vaccine confidence, followed closely by public education intervention. Actively training front line providers about vaccines (especially new vaccines such as the COVID vaccine and the HPV vaccine) remains a high impact activity that is likely to improve vaccine confidence overall. Fortunately, community-level healthcare practitioners do trust the government health authorities since they are typically the first stop for new healthcare information. Second, public education intervention appears to be effective in increasing knowledge and vaccine confidence. Maintaining the vaccine confidence of healthcare providers will clearly have an impact on vaccine confidence among these individuals, and that is also likely to have a positive impact on vaccine uptake in the general population. Furthermore, as Mali prepares for HPV vaccination campaigns, an investment in vaccine education prior to the mass vaccination campaign will surely have a significant impact on future generations of the people of Mali. Our plan is to evaluate additional aspects of vaccine confidence that have not previously been addressed in LMIC such as access to vaccines (distance and availability) and healthcare (for lower income individuals) as determinants of vaccination rates. We will also continue to monitor the erosion of vaccine confidence in Mali and the impact of vaccine misinformation on uptake of new vaccines.

Comment: There appears to be a discrepancy between the title and the content of the paper. It may be advisable to modify the title to better reflect the study’s focus for clarity. Additionally, it is recommended to adjust the citation format (reference format) in accordance with the submission guidelines.

Reply: Thank you for the request for clarification. The title is:

Vaccine Confidence during Public Health Challenges and Prior to HPV Vaccine Introduction in Mali

As described above, there was a unique juxtaposition of the introduction of a novel vaccine COVID), and potentially a second vaccine (HPV) in the time period under study. At the time, there were also multiple challenges to the proper management of public health. These were: (1) A military coup; (2) the COVID pandemic.

We have experimented with several versions of the title and find it difficult to describe the study, since all of those factors were in play at the same time. None of the alternative versions of the title describe the manuscript as succinctly as this one.

If the reviewer has an alternative suggestion, we are happy to consider it.

If not, then we respectfully request to have the title stay unchanged and we hope that the clarifications made to the Introduction of the Manuscript, to better enumerate the challenges, will satisfy the reviewer.

Reviewer 2 Report

Comments and Suggestions for Authors This study, entitled "Vaccine Confidence during Public Health Challenges and Prior to HPV Vaccine Introduction in Mali", addresses an important and under-researched topic. It provides valuable insights for low- and middle-income countries, particularly in relation to how confidence in healthcare providers versus public authorities affects vaccine introduction. The mixed methods approach (surveys, public education, training of healthcare providers) is well integrated and appropriate for the objectives. However, the originality of the study would be strengthened by more comparative analyses with similar LMIC contexts. Some sentences are too long or too complex to be comprehensible, especially in the methods and discussion, and the authors use several acronyms such as “HCP” without defining them on first use. More details about possible biases (e.g., self-selection or interviewer influence) would be desirable. Discuss more explicitly how these results can be generalised to other LMICs or environments with political instability. Some figures need to be improved, such as Figure 1 Comments on the Quality of English Language

The English could be improved to more clearly express the research.

Author Response

Reviewer 2

Comment: This study, entitled "Vaccine Confidence during Public Health Challenges and Prior to HPV Vaccine Introduction in Mali", addresses an important and under-researched topic. It provides valuable insights for low- and middle-income countries, particularly in relation to how confidence in healthcare providers versus public authorities affects vaccine introduction. The mixed methods approach (surveys, public education, training of healthcare providers) is well integrated and appropriate for the objectives. However, the originality of the study would be strengthened by more comparative analyses with similar LMIC contexts. Some sentences are too long or too complex to be comprehensible, especially in the methods and discussion, and the authors use several acronyms such as “HCP” without defining them on first use. More details about possible biases (e.g., self-selection or interviewer influence) would be desirable. Discuss more explicitly how these results can be generalised to other LMICs or environments with political instability. Some figures need to be improved, such as Figure 1.

Point by point: 

Comment: The originality of the study would be strengthened by more comparative analyses with similar LMIC contexts.

Reply:

Thank you for this suggestion. We have included many more references to other studies as there are quite a few. More than 10 references were added. Thank you for the opportunity to point to other important studies in the field. See for example this section:

Compared to other studies of vaccine confidence, this study confirms that trusting the national public health agency would be strongly correlated with several vaccine outcomes (Arsenault et al., 2025) . We also note that other studies evaluated healthcare ‘connectedness’ as a measure of vaccine confidence. For example, having preventive health services in the last year was associated with COVID-19 vaccination in Italy, Mexico, the United Kingdom (UK), and the United States (US) and with vaccine confidence in the US (RR 1.10) ) (Arsenault et al., 2025). Similar to the findings in our study, ‘trust in healthcare provider” and health system engagement was associated with routine vaccinations (although not necessarily with COVID-19 vaccination) and with vaccine uptake (Demi et al., 2022).

The timing of, and access to vaccination, may also be factors: in many countries HCP were prioritized and in some parts of the USA COVID vaccination was mandated resulting in high vaccination rates among HCP. The same may also have been true in Mali, as HCP were also prioritized as soon as vaccines became available. Access to medical centers has been associated with higher HPV vaccination rates (Oka et al, 2025). A review of vaccine confidence studies showed that little information is available on ‘convenience’ and this should be included in future studies of vaccine confidence. For example, it is not clear whether ‘access to vaccines’ was responsible for the higher rate of vaccination among males. This has been observed in Kenya as well (Njororai et al., 2023) .

As with location of the vaccination program, integration of the program into routine vaccination is likely to be a key factor that improves vaccine acceptance (Xu et al., 2024) . Notably, Mali decided against providing HPV vaccine at school and to both genders, factors that may decrease HPV vaccine uptake. Instead, the vaccine will be provided as a single dose and as part of the routine childhood vaccination schedule, factors have been associated with increased vaccine uptake (Xu et al., 202; Robles et al. 2018).

The survey results also showed that many well-known misinformation themes had emerged based on videos that were widely circulated on WhatsApp and other social media platforms. Misinformation has also been reported to have a negative effect on COVID-19 prevention measures in other LMIC (low to middle-income countries) in West Africa such as Cameroon and Ghana. There was no correlation in this study between education-level and belief in common rumors/misinformation about the COVID-19 vaccine suggesting that the well-educated were equally exposed to misinformation as those with no formal schooling.

Comment: Some sentences are too long or too complex to be comprehensible, especially in the methods and discussion

Reply: Thank you for the suggestion.

We have reviewed the MS for long sentences (compound sentences) and have modified them to clarify the manuscript. 

Comment: The authors use several acronyms such as “HCP” without defining them on first use.

Reply:.

Thank you for this request for clarification We have checked to make sure that the acronyms are spelled out the first time that they are used.

Comment: More details about possible biases (e.g., self-selection or interviewer influence) would be desirable.

Reply:

This is an excellent suggestion, and we apologize for overlooking this aspect of the manuscript. We have included a new section on possible biases, for example, self-selection or interviewer influence).

As with all research, there are several limitations to this study that should be improved upon in future research. First, all interviews were conducted by healthcare providers at community clinics. While this strategy was convenient and helped ensure that community members were comfortable, it may have introduced bias when community members were asked to identify whom they trusted for advice. Future projects should work to identify surveyors who are a part of the community, but not necessarily healthcare providers themselves.

Second, this study was conducted in the poorest commune in Bamako and one rural site without the inclusion of any more wealthy areas for comparison. Future studies would benefit from including a more well educated and wealthy area for comparison. Finally, the survey for this project proved to be much too long as we found many participants would stop answering after the first two pages. Limiting the number of questions is important to ensure the participant really gives themselves time to answer thoughtfully.

Comment: Discuss more explicitly how these results can be generalised to other LMICs or environments with political instability.

Reply:

Thank you for this suggestion. We have addressed this issue as requested in the Discussion.

... Discussions with their local healthcare providers were the single most effective element of this campaign at increasing vaccine confidence, followed closely by public education intervention. Actively training front line providers about vaccines (especially new vaccines such as the COVID vaccine and the HPV vaccine) remains a high impact activity that is likely to improve vaccine confidence overall. Fortunately, community-level healthcare practitioners do trust the government health authorities since they are typically the first stop for new healthcare information. Second, public education intervention appears to be effective in increasing knowledge and vaccine confidence. Maintaining the vaccine confidence of healthcare providers will clearly have an impact on vaccine confidence among these individuals, and that is also likely to have a positive impact on vaccine uptake in the general population. Furthermore, as Mali prepares for HPV vaccination campaigns, an investment in vaccine education prior to the mass vaccination campaign will surely have a significant impact on future generations of the people of Mali.

Our plan is to evaluate additional aspects of vaccine confidence that have not previously been addressed in LMIC such as access to vaccines (distance and availability) and healthcare (for lower income individuals) as determinants of vaccination rates. We will also continue to monitor the erosion of vaccine confidence in Mali and the impact of vaccine misinformation on uptake of new vaccines.

Comment: Some figures need to be improved, such as Figure 1

Reply: Figure 1 was replaced with a full image of the story-telling cloth and a caption was added.

Reviewer 3 Report

Comments and Suggestions for Authors

This timely and well-structured communication article addresses a critical public health issue in Mali — the challenge of building and maintaining vaccine confidence in the context of both the COVID-19 pandemic and political instability, especially preceding the rollout of the HPV vaccine. The manuscript provides novel insights using community-based surveys, healthcare provider (HCP) training, and a multi-component public health intervention. The work is particularly valuable due to its context-specific approach and could serve as a model for similar low- and middle-income country (LMIC) settings.

Some points for improvement:

  1.  Authors need to differentiate between the objectives related to COVID-19 vaccine confidence and those connected to informing the future HPV campaign. The introduction currently blends the two, which could be slightly confusing to the reader. Please clarify the study objectives and design.
  2. Methods. Please clarify handling of missing data (especially in the HCP pre/post surveys).

  3. The full version of the questionnaire used should be included in the supplementary materials.
  4. Please provide more detailed information on who developed the questionnaire, how it was used and distributed, and how its validation was conducted.
  5. Table 2. P-values are noted, but confidence intervals for key odds ratios would improve interpretability.

  6. The paper would benefit from a clearly labeled limitations section.
  7. The title and abstract emphasize the relevance to HPV vaccine rollout, but the actual HPV-related findings are sparse. Suggest including a short forward-looking section or paragraph on how these lessons will shape the HPV campaign, with more specifics if possible.

  8. Line 296–300: The sentence “Discussions with their local healthcare providers was the single most effective element...” → should be “were” instead of “was” (subject–verb agreement).

Author Response

Reviewer 3

This timely and well-structured communication article addresses a critical public health issue in Mali — the challenge of building and maintaining vaccine confidence in the context of both the COVID-19 pandemic and political instability, especially preceding the rollout of the HPV vaccine. The manuscript provides novel insights using community-based surveys, healthcare provider (HCP) training, and a multi-component public health intervention. The work is particularly valuable due to its context-specific approach and could serve as a model for similar low- and middle-income country (LMIC) settings.

Some points for improvement:

Comment: Authors need to differentiate between the objectives related to COVID-19 vaccine confidence and those connected to informing the future HPV campaign. The introduction currently blends the two, which could be slightly confusing to the reader. Please clarify the study objectives and design.

Reply:

Thank you for the suggestions for clarification. We have added clarification to the sections relating to COVID vaccine and HPV in the manuscript.

Comment: Methods. Please clarify handling of missing data (especially in the HCP pre/post surveys).

Reply:

Thank you for this comment. We added the text below to the methods section.

Missing data were dropped from all analyses where that question/data point was included. When evaluating the pre/post surveys for HCP, the HCP would be dropped from the analysis entirely if they did not answer the question being evaluated.

Comment: The full version of the questionnaire used should be included in the supplementary materials.

Reply:

This was an oversight and it is now included.

Comment: Please provide more detailed information on who developed the questionnaire, how it was used and distributed, and how its validation was conducted.

Reply:

The supplemental materials include detailed information about the drafting process of the surveys. We have made that clearer in the MS.

Comment: Table 2. P-values are noted, but confidence intervals for key odds ratios would improve interpretability.

Reply: 

Confidence intervals for key odds ratios were added in brackets under standard errors in Tables 2 and 3.

Comment: The paper would benefit from a clearly labeled limitations section.

Reply:

This is an excellent suggestion and was also requested by another reviewer. We have now included a section on potential biases and limitations. (Starting at line 404).

As with all research, there are several limitations to this study that should be improved upon in future research. First, all interviews were conducted by healthcare providers at community clinics. While this strategy was convenient and helped ensure that community members were comfortable, it may have introduced bias when community members were asked to identify whom they trusted for advice. Future projects should work to identify surveyors who are a part of the community, but not necessarily healthcare providers themselves.

Second, this study was conducted in the poorest commune in Bamako and one rural site without the inclusion of any more wealthy areas for comparison. Future studies would benefit from including a more well educated and wealthy area for comparison. Finally, the survey for this project proved to be much too long as we found many participants would stop answering after the first two pages. Limiting the number of questions is important to ensure the participant really gives themselves time to answer thoughtfully.

Comment: The title and abstract emphasize the relevance to HPV vaccine rollout, but the actual HPV-related findings are sparse. Suggest including a short forward-looking section or paragraph on how these lessons will shape the HPV campaign, with more specifics if possible.

Reply: Thank you for the suggestion.

We have also now included a short forward-looking section that addresses how we are integrating this information into our current HPV vaccine campaign activities. (Starting at 417).

... We plan to use the “story telling cloth” approach to support the HPV vaccine roll-out in Mali that is ongoing. We will also evaluate aspects of vaccine confidence that have not previously been addressed in LMIC such as access to vaccines (distance and availability) and healthcare (for lower income individuals) as determinants of vaccination rates. We will continue to monitor the erosion of vaccine confidence in Mali and the impact of vaccine misinformation on uptake of new vaccines.

Comment: Line 296–300: The sentence “Discussions with their local healthcare providers was the single most effective element...” → should be “were” instead of “was” (subject–verb agreement).

Reply:

Thank you for finding this error, we have corrected the subject-verb agreement.

Round 2

Reviewer 1 Report

Comments and Suggestions for Authors

Your response appropriately addresses my review comments. Clarifying the introduction of the manuscript has improved the quality of the manuscript.

Author Response

Reviewer 1: Your response appropriately addresses my review comments. Clarifying the introduction of the manuscript has improved the quality of the manuscript.

Response: Thank you very much.  Your suggestions were very helpful. 

Reviewer 2 Report

Comments and Suggestions for Authors I thank you for your thoughtful responses to the reviewers' comments and for the revisions to the manuscript. I recognise the efforts made to address the points raised and to improve the overall clarity and quality of the text. However, I would like to point out that there are still several instances where acronyms such as HPV, HIV and others appear without being spelt out at the first mention in the main text. For the sake of clarity and consistency, especially for readers who are not familiar with all acronyms, I recommend defining all acronyms the first time they are used in the manuscript. Once this issue is resolved, I believe the manuscript will be suitable for publication.

Author Response

Reviewer 2: I thank you for your thoughtful responses to the reviewers' comments and for the revisions to the manuscript. I recognise the efforts made to address the points raised and to improve the overall clarity and quality of the text. However, I would like to point out that there are still several instances where acronyms such as HPV, HIV and others appear without being spelt out at the first mention in the main text. For the sake of clarity and consistency, especially for readers who are not familiar with all acronyms, I recommend defining all acronyms the first time they are used in the manuscript. Once this issue is resolved, I believe the manuscript will be suitable for publication.

Reply: Thank you for pointing out that we did not explain HIV and HPV (and perhaps other terms). We will review the manuscript again and will address this oversight. Thank you for pointing it out and the revised manuscript now should not have unexplained acronyms.